# Cosmetic Application of Cyanobacteria Extracts with a Sustainable Vision to Skincare: Role in the Antioxidant and Antiaging Process

**DOI:** 10.3390/md20120761

**Published:** 2022-12-02

**Authors:** Janaína Morone, Graciliana Lopes, João Morais, Jorge Neves, Vítor Vasconcelos, Rosário Martins

**Affiliations:** 1CIIMAR/CIMAR—Interdisciplinary Centre of Marine and Environmental Research, University of Porto, Terminal de Cruzeiros do Porto de Leixões, Av. General Norton de Matos s/n, 4450-208 Matosinhos, Portugal; 2FCUP—Faculty of Sciences, University of Porto, Rua do Campo Alegre, Edifício FC4, 4169-007 Porto, Portugal; 3Health and Environment Research Centre, School of Health, Polytechnic Institute of Porto, Rua Dr. António Bernardino de Almeida, 400, 4200-072 Porto, Portugal

**Keywords:** cyanobacteria, cosmetics, carotenoids, phycobiliproteins, phenols, metalloproteinases, oxidative stress, antiaging

## Abstract

Nature-based and sustainably sourced cosmetics have been dominating the area of skincare products worldwide. Due to their antioxidant and antiaging properties, compounds from cyanobacteria, such as carotenoids and phycobiliproteins, may replace synthetic ingredients in cosmetic formulations and may be used in products such as sunscreens, skincare creams, and makeup. In this study, we evaluated the potential of acetonic and aqueous extracts from cyanobacteria strains of the genera *Cyanobium* and *Leptothoe* and from strains within Synechococcales and Oscillatoriales orders, for use in cosmetics. Extractions were sequentially performed with acetone and water. Extracts were firstly analyzed for their toxicity to keratinocytes, fibroblasts, and endothelial cells (HaCAT, 3T3L1 and hCMEC/D3, respectively). The non-cytotoxic extracts were characterized in terms of total proteins, carotenoids, chlorophyll, phenols, phycobiliproteins, and analyzed for their antioxidant potential against the superoxide anion radical (O_2_^•−^), and for their ability to inhibit key enzymes associated with the skin aging process. Aqueous extracts were richer in total proteins and phycobiliproteins. The aqueous extracts of Synechococcales cyanobacterium LEGE 181157 and Synechococcales cyanobacterium LEGE 181150 showed the highest value for total proteins (760.81 and 695.25 μg BSA mL^−1^_dry extract,_ respectively) and the best values regarding O_2_^•−^ scavenging (IC_50_ = 63.24 and 112.18 μg mL^−1^_dry extract_, respectively) with a significant negative correlation observed (*p* < 0.01). Moreover, aqueous extracts of Synechococcales cyanobacterium LEGE 181150 and Synechococcales cyanobacterium LEGE 181157 inhibited hyaluronidase, (IC_50_ of 483.86 and 645.06 μg mL^−1^_dry extract_, respectively), with a significant negative correlation with total proteins (*p* < 0.05), pointing out the contribution of these compounds to the biological activities observed. Acetonic extracts were richer in carotenoids and phenols. Zeaxanthin and β-carotene were predominant among all strains, being present in higher amount in *Cyanobium* sp. LEGE 07175 (53.08 μg mg^−1^) and *Leptothoe* sp. LEGE 181156 (47.89 μg mg^−1^), respectively. The same strains also showed the highest values for collagenase inhibition at 750 μg mL^−1^_dry extract_ (32.88 and 36.61%, respectively). Furthermore, *Leptothoe* sp. LEGE 181156 exhibited the lowest IC_50_ value for tyrosinase inhibition (465.92 μg mL^−1^_dry extract_) and Synechococcales cyanobacterium LEGE 181157 presented the best values for elastase inhibition (IC_50_ of 380.50 and IC_25_ of 51.43 μg mL^−1^_dry extract_). In general, cyanobacteria extracts demonstrated potential for being used for antiaging purposes, with aqueous extracts being more efficient at free radicals scavenging and acetonic ones at avoiding degradation of dermal matrix components.

## 1. Introduction

Thriving cosmetics properties include skin protection against chemical and physical factors such as xenobiotics, ultraviolet radiation (UVR), and dehydration, the main factors involved in skin’s loss of integrity and aging. Despite being a result of natural physiological features, the skin aging process may be accelerated by mechanisms such as oxidative stress, which settles down with the accumulation of free radicals. Oxidative stress is mainly responsible for triggering the activation of matrix metalloproteinases (MMPs), namely collagenases, elastases, and hyaluronidases, that contribute to the degradation of collagen, elastin, and hyaluronic acid (HA), as the main components of the extracellular matrix (ECM), which are essential for the maintenance of skin elasticity, firmness, and regeneration [1]. Consequently, the aesthetic appearance of the skin becomes compromised, which affects health, social well-being, and self-esteem. Thus, skincare habits have become a priority among people who seek a healthier and more pleasing lifestyle.

Cosmetic ingredients derived from photosynthetic organisms are a beneficial approach to prioritize the use of natural raw materials into final products that respect the skin’s properties and the environment and, consequently, make a commitment to sustainability [2]. Among photosynthetic organisms, the interest in cyanobacteria has been growing, with different species being explored from the natural cosmetics point of view. Cyanobacteria are widespread organisms found in all ecosystems, some with life-threatening conditions such as high temperatures, high salinity, and extreme solar radiation [3]. As extremely adaptable organisms, cyanobacteria have developed a remarkable secondary metabolism, resulting in an arsenal of metabolites with proven pharmaceutical, nutraceutical, and food-conservation potential. Considering skin-related needs, the production of carotenoids, phenolic compounds, phycobiliproteins (PBPs), mycosporin-like amino acids (MAAs), and scytonemin (SCY) places these organisms at the forefront of healthy skincare. All these molecules play a central role as antiaging ingredients, mainly due to their antioxidant potential, sun-protection capacity, and ability to inhibit enzymes responsible for the degradation of the ECM [2,4].

The use of cyanobacteria-derived compounds in skincare products can be expressed in sunscreens, moisturizing creams, formulations with antioxidant and anti-inflammatory properties, as well as in pigmentation disorders, dermatophytosis, and infectious diseases [1,2,5,6,7]. Recent studies focusing on the potential of cyanobacteria in cosmetics have reported the antioxidant activity attributed to the production of carotenoids from some genera, namely *Leptolyngbya, Synechocystis,* and *Wollea* [2,8], PBP such as phycocyanin (PC) and phycoerythrin (PE) from *Arthrospira* sp. and *Spirulina* sp. [8,9], and phenolic compounds from *Nostoc commune* [10]. Considering the inhibition of MMPs compounds such as mycosporin-2-glycine isolated from a halotolerant cyanobacterium was found to inhibit collagenase activity [11], cyclodepsipeptide from *Schizothrix* sp. and *Coleofasciculus* sp. presented high elastase inhibitory properties [12], and a polyssacharide from *Nostochopsis lobatus* showed potent hyaluronidase inhibitory activity [13]. Additionally, Favas and co-workers recently showed that some cyanobacteria aqueous extracts had the potential to retard dermal density loss [14]. Additionally, the extract from *Phormidium persicinum* (PHORMISKIN Bioprotech G^®^) has been commercially used as a melanin synthesis reducer [15]. The studies involving cyanobacteria as photoprotectors have also been highlighted [16]. It is known that cyanobacteria synthetize pigments, which are UVR-absorbing compounds, namely carotenoids, PBPs, MAAs, and SCY [16]. The available literature produced so far highlights the biotechnological potential of cyanobacteria as natural and eco-friendly sources of compounds in the field of skincare and triggers further exploitation in this field. Besides being self-renewable, with low nutritional requirements, and presenting high growth and photosynthesis rates, the sustainable cultivation of cyanobacteria is guaranteed [2,17].

In the present work, seven cyanobacteria strains isolated from Portuguese and Cape Verde marine ecosystems (*Cyanobium* sp. LEGE 07175, Synechococcales cyanobacterium LEGE 181150, *Leptothoe* sp. LEGE 181155, *Leptothoe* sp. LEGE 181156, Synechococcales cyanobacterium LEGE 181157, Synechococcales cyanobacterium LEGE 181158, and Oscillatoriales cyanobacterium LEGE 181159) were explored for their potential to slow down skin aging. Acetonic and aqueous extracts were evaluated for cytotoxicity in skin cell-lines, phytochemical composition, antioxidant potential, and inhibitory effect in skin key enzymes. 

## 2. Results

### 2.1. Cytotoxicity

Cytotoxic effects were registered in all cell lines for both acetonic and aqueous extracts of the strain *Leptothoe* sp. LEGE 181155 (g and n in Appendix A) and for the acetonic extract of strain Oscillatoriales cyanobacterium LEGE 181159 (m in Appendix A), with a significant decrease from the lowest concentration (12.5 μg mL^−1^) (*p* < 0.05) after 24 h For all other strains, none of the extracts showed cytotoxic effects, with no significant differences from the control (*p* < 0.05) for both 24 and 48 h of exposition. Based on these results, the extracts of the strain *Leptothoe* sp. LEGE 181155 and the acetone extract of Oscillatoriales cyanobacterium LEGE 181159 were excluded from the study (Results and Appendix A).

### 2.2. Phytochemical Analysis

For the purpose of comparing the chemical profiles and establishing a relationship between the chemical composition and the biological activities assessed herein, the acetonic and aqueous extracts were characterized in terms of phenolic content, total proteins, and PBP. Moreover, the qualitative and quantitative carotenoids profile was also established.

#### 2.2.1. Carotenoids

The HPLC-PDA analysis of the acetonic extract from the different cyanobacteria strains allowed the detection of 13 carotenoids, 2 chlorophyll-*a* derivatives, and chlorophyll-*a,* and for the aqueous extract 8 carotenoids, 2 chlorophyll-*a* derivatives and chlorophyll-*a* (Table 1). The chromatographic profile of the three different genera is illustrated in Figure 1.

The identified compounds for aqueous extracts consisted of two xanthophylls: zeaxanthin (**5**) and myxoxanthophyll (**7**); one carotene: β-carotene (**13**), chlorophyll-*a* (**4**) and the chlorophyll-*a* derivative phaeophytin-*a* (**14**). For acetonic extracts, the identified compounds consisted of four xanthophylls: lutein (**2**), zeaxanthin (**5**), myxoxanthophyll (**7**) and echinenone (**10**); two carotenes: β-carotene (**13**) and γ-carotene (**16**), as well as chlorophyll-*a* (**4**) and phaeophytin-*a* (**14**). Seven compounds with the same spectra as the identified carotenoids, however, with different retention times to the standard, were also detected in some strains, being defined as unidentified carotenoids (**1**, **3**, **8**, **9**, **11**, **12**, **15**), and the same was carried out for the chlorophyll-*a* derivative (**6**). Regarding aqueous extracts, five compounds were defined as unidentified carotenoids (**1**, **3**, **9**, **12**, **15**). 

For aqueous extracts, total carotenoid concentration ranged between 0.05 and 3.00 μg mg^−1^ of dry extract. *Cyanobium* sp. LEGE 07175 showed the highest carotenoids content, followed by Synechococcales cyanobacterium LEGE 181150 and Oscillatoriales cyanobacterium LEGE 181159 (3.00; 1.55 and 0.84 μg mg^−1^, respectively) (*p* < 0.05). All strains contain β-carotene (**13**), Zeaxanthin (**5**), and phaeophytin-*a* (**14**), but in *Cyanobium* sp. LEGE 07175, this chlorophyll-*a* derivative could not be quantified. Myxoxanthophyll (**7**) was not found in two of six strains. Lutein (**2**) and γ-carotene (**16**) were not identified in any aqueous extract.

According to chlorophylls, the total chlorophylls content ranged from 0.81 up to 14.77 μg mg^−1^. Chlorophyll-*a* (**4**) was found in all strains, except in *Cyanobium* sp. LEGE 07175, where only a chlorophyll-*a* derivative (**6**) was found. The highest chlorophylls content was found in Synechococcales cyanobacterium LEGE 181150 (14.77 μg mg^−1^) (*p* < 0.05), followed by Synechococcales cyanobacterium LEGE 181158 and Oscillatoriales cyanobacterium LEGE 181159 (9.42 and 8.69 μg mg^−1^), with no significant differences.

Regarding acetonic extract, the total carotenoid concentration ranged between 104.71 and 150.15 μg mg^−1^ of dry extract, and it differed significantly. The highest carotenoids content was found in *Leptothoe* sp. LEGE 181156 followed by Synechococcales cyanobacterium LEGE 181150 and *Cyanobium* sp. LEGE 07175 (150.15; 123.26; and 115.98 μg mg^−1^, respectively) (*p* < 0.05). Lutein (**2**) was found in two strains: Synechococcales cyanobacterium LEGE 181157 and Synechococcales cyanobacterium LEGE 181158. Zeaxanthin (**5**) was identified in all strains, but in Synechococcales cyanobacterium LEGE 181150 and Synechococcales cyanobacterium LEGE 181158, this xanthophyll appeared co-eluted with chlorophyll-*a* (**4**). The highest content of this xanthophyll was found in *Cyanobium* sp. LEGE 07175 (53.08 μg mg^−1^) (*p* < 0.05). Even though *Cyanobium* sp. LEGE 07175 was the only one that did not present myxoxanthophyll (**7**), it was the only strain in which echinenone (**10**) was identified. All the other strains presented β-carotene (**13**), phaeophytin-*a* (**14**), and γ-carotene (**16**) in their profile. The highest content of β-carotene was found in the strain *Leptothoe* sp. LEGE 181156 (47.89 μg mg^−1^) (*p* < 0.05) followed by Synechococcales cyanobacterium LEGE 181158 and Synechococcales cyanobacterium LEGE 181157 (30.57 and 29.30 μg mg^−1^, respectively). The highest phaeophytin-*a* content was detected in Synechococcales cyanobacterium LEGE 181158, followed by *Cyanobium* sp. LEGE 07175 and Synechococcales cyanobacterium LEGE 181157 (6.31; 5.81; and 5.23 μg mg^−1^, respectively) with no significant differences (Table 1 and Figure 1).

Regarding chlorophylls, the total chlorophylls content ranged from 12.69 up to 34.81 μg mg^−1^. Chlorophyll-*a* (**4**) derivative was identified in all strains of acetonic extracts in considerably high amounts when compared with chlorophyll-*a*. Synechoccocales cyanobacterium LEGE 181156 presented the highest content, followed by Synechococcales cyanobacterium LEGE 181150 and Synechococcales cyanobacterium LEGE 181158 (34.81; 29.23 and 21.90 μg mg^−1^, respectively) with no significant differences. Additionally, in two strains, namely Synechococcales cyanobacterium LEGE 181150 and Synechococcales cyanobacterium LEGE 181158, chlorophyll-*a* (**4**) was also present; however, it was co-eluted with zeaxanthin (**5**). Amongst the pigments analyzed, carotenoids were dominant over chlorophylls in all strains for acetonic extraction. Furthermore, zeaxanthin, phaeophytin-*a,* and γ-carotene were identified in the acetonic extracts of all the species studied (Figure 1, Table 1).

Acetonic extracts were richer in pigments than aqueous extracts, considering both carotenoids and chlorophylls. The unique carotenoid identified for all strains, in both extractions, was β-carotene. Moreover, the carotene echinenone was only identified in *Cyanobium* sp. LEGE 07175.

#### 2.2.2. Total Phenolic Content (TPC)

For both extracts, TPC was measured through the Folin–Ciocalteu colorimetric assay; the results were expressed in μg of GAEs per mg of dry extract and are displayed in Table 2.

Acetonic extracts presented higher TPC values than aqueous extracts. The highest TPC was found in the acetonic extract of *Leptothoe* sp. LEGE 181156, followed by Synechococcales cyanobacterium LEGE 181158 and *Cyanobium* sp. LEGE 07175 (27.13; 24.71; and 23.11 μg GAE mg^−1^_dry extract_, respectively) with not significant differences.

Regarding aqueous extracts, the values ranged from 11.46 up to 15.67 μg GAE mg^−1^_dry extract_, Synechococcales cyanobacterium LEGE 181158 being the strain with the highest TPC, with no significant differences among the samples, except between Oscillatoriales cyanobacterium LEGE 181159, which showed lower content, and Synechococcales cyanobacterium LEGE 181158 (Table 2).

#### 2.2.3. Total Proteins

The total protein content of both extracts was evaluated by the method of BSA, using the protein assay kit, and the results were expressed in μg of BSA per mg of dry extract (Table 2). Higher values were found in the aqueous extraction, in which Synechococcales cyanobacterium LEGE 181157 presented the highest content, followed by Synechococcales cyanobacterium LEGE 181150 and Synechococcales cyanobacterium LEGE 181158 (760.81; 695.25 and 579.99 μg BSA mg^−1^_dry extract_, respectively) (*p* < 0.05). 

Regarding acetonic extracts, Synechococcales cyanobacterium LEGE 181158 showed the highest content, followed by Synechococcales cyanobacterium LEGE 181150 (209.31 and 199.37 μg BSA mg^−1^_dry extract_, respectively) with no significant difference.

#### 2.2.4. Phycobiliproteins (PBPs)

As water-soluble proteins, PBPs were quantified only in aqueous extracts. Regarding PC, the Synechococcales cyanobacterium LEGE 181150 extract presented the highest content, followed by Oscillatoriales cyanobacterium LEGE 181159 and Synechococcales cyanobacterium LEGE 181158 (222.76; 201.15; and 194.43 μg mg^−1^_dry extract_, respectively) (*p* < 0.05). In contrast, Synechococcales cyanobacterium LEGE 181157 possessed the highest content of PE, followed by *Leptothoe* sp. LEGE 181156 and Oscillatoriales cyanobacterium LEGE 181159 (275.04; 159.05 and 26.33 μg mg^−1^_dry extract,_ respectively) (*p* < 0.05) (Table 3).

The strong blue color of PC was notable is the aqueous extracts of Synechococcales cyanobacterium LEGE 181150, Oscillatoriales cyanobacterium LEGE 181159, and Synechococcales cyanobacterium LEGE 181158; and the strong pink/red color of PE was observed in the aqueous extracts of Synechococcales cyanobacterium LEGE 181157 and *Leptothoe* sp. LEGE 181156, which is in accordance with the main phycobiliproteins quantified in each strain.

### 2.3. Biological Activities

Superoxide Anion Radical (O_2_^•−^) Scavenging

The results of O_2_^•−^ scavening capacity of both cyanobacteria extracts are summarized in Table 4 and Figure 2. Aqueous extracts were more effective and presented lower IC_50_ values than acetonic extracts towards radical scavenging. Synechococcales cyanobacterium LEGE 181157 was the most effective strain, showing the lowest IC_50_ value (63.24 μg mL^−1^_dry extract_), followed by Synechococcales cyanobacterium LEGE 181150 (112.18 μg mL^−1^_dry extract_), with no significant difference, and Oscillatoriales cyanobacterium LEGE 181159 (332.40 μg mL^−1^dry extract) (*p* < 0.05).

Concerning acetonic extracts, the lowest IC_50_ was presented by Synechococcales cyanobacterium LEGE 181157, followed by Synechococcales cyanobacterium LEGE 181150 and Leptothoe sp. LEGE 181156 (847.65; 943.45; and 1097.05 μg mL^−1^_dry extract_) with no significant differences among the species (Table 4).

### 2.4. Enzymes Inhibition

#### 2.4.1. Hyaluronidase (Hase) Inhibition

For both extracts studied herein, the strains of the order Synechoccales presented higher potential for Hase inhibition. Synechococcales cyanobacterium LEGE 181150 showed the lowest IC_50_ value for aqueous and acetonic extracts (483.86 and 726.29 μg mL^−1^_dry extract_, respectively), followed by Synechococcales cyanobacterium LEGE 181158 (624.51 and 738.88 μg mL^−1^_dry extract_, respectively) and Synechococcales cyanobacterium LEGE 181157 (645.06 and 859.83 μg mL^−1^_dry extract_, respectively), with no significant differences between them. Leptothoe sp. LEGE 181156 was the unique strain for which the acetonic extracts did not reach IC_50_ nor IC_25_ values (Table 5).

#### 2.4.2. Elastase Inhibition

For elastase, only the acetonic extracts showed inhibitory potential, with Synechococcales cyanobacterium LEGE 181157 being the only strain that reached IC_50_, with a value of 380.50 μg mL^−1^_dry extract_. Regarding IC_25_, this strain reached an interesting value (51.43 μg mL^−1^_dry extract_) (*p* < 0.05), followed by Synechococcales cyanobacterium LEGE 181158 and *Leptothoe* sp. LEGE 181156 (233.47 and 378.26 μg mL^−1^_dry extract_) (Table 5).

#### 2.4.3. Tyrosinase Inhibition

As for elastase, only acetonic extracts were able to promote tyrosinase inhibition (Table 5). *Leptothoe* sp. LEGE 181156 reached IC_50_, followed by Synechococcales cyanobacterium LEGE 181157 (465.92 and 849.48 mg mL^−1^, respectively) (*p* < 0.05). Additionally, both were also able to reach IC_25_ with values of 133.47 and 381.33 mg mL^−1^, respectively) with no significant differences.

#### 2.4.4. Collagenase Inhibition

Contrary to the previous enzymes, the IC values were not able to be determined for collagenase. Due to the scarce amount of enzyme available, we chose to carry out the assay in kinetic mode, using the result of the calculation by slope of the linear equation line. In this sense, the inhibitory activity of the extracts was tested at two different concentrations of 0.75 and 1 mg mL^−1^.

Acetonic extracts were more effective than aqueous ones towards collagenase inhibition (Table 6). *Leptothoe* sp. LEGE 181156 presented 36.61% of collagenase inhibition for the tested concentration of 0.75 mg mL^−1^, while *Cyanobium* sp. LEGE 07175 reached 32.88% for the same concentration. Although Synechococcales cyanobacterium LEGE 181150 showed 9.81% of collagenase inhibition at 0.75 mg mL^−1^, this strain presented 44.40% when tested at 1.0 mg mL^−1^.

Regarding aqueous extracts, Synechococcales cyanobacterium LEGE 181158 was the most effective strain against collagenase, presenting 25.97% of inhibition at 1.0 mg mL^−1^.

## 3. Discussion

The stratified squamous epithelium that constitutes the epidermis is mainly composed of keratinocytes and corneocytes and is in a constant process of cell renewal. In the dermis, fibroblasts are essential cells involved in the production and maintenance of the collagen and elastic fibers that ensure resistance and the elasticity of the skin. In addition, dermis is richly vascularized, being responsible for the nutrition and oxygenation of the epidermis [1].The absence of cytotoxicity to main skin cells, both from the epidermis and dermis, is mandatory in a skin formulation. In this work, although seven strains were initially selected, the strain Leptothoe sp. LEGE 181155 was excluded due to the cytotoxic effects of both extracts on keratinocytes, fibroblasts, and endothelial cells. Additionally, due to cytotoxic effects in all cell lines tested, the acetone extract of Oscillatoriales cyanobacterium LEGE 181159 was also excluded. Thus, only nontoxic extracts were further analyzed.

### 3.1. Phytochemical Analysis

#### 3.1.1. Carotenoids and Chlorophylls

Among the studied strains, the acetonic extract of the strain *Leptothoe* sp. LEGE 181156 was the richest in carotenoids (150.15 μg mg^−1^ _dry extract_) (*p* < 0.05). The highest concentration of carotenoids in aqueous extracts was obtained with strain *Cyanobium* sp. LEGE 07175 with a maximum of 3.00 μg mL^−1^ _dry extract_. As expected, the results obtained indicated that acetone extraction is more efficient than the aqueous on both carotenoids and chlorophylls (Table 1), which can be explained by the lower polar nature of these compounds, and therefore a higher affinity for lower polar solvents such as acetone. Additionally, as extractions were carried out in sequential mode, in which the acetonic extraction was performed first, most of these pigments were extracted with acetone.

Considering carotenoids identification, the xanthophyll zeaxanthin was found in all acetonic extracts of all species studied, ranging from 24.38 to 53.08 μg mg^−1^_dry extract_, with the highest value reported in *Cyanobium* sp. LEGE 07175, followed by *Leptothoe* sp. LEGE 181156 (53.08 and 39.85 µg mg^−1^ _dry extract_, respectively) (*p* < 0.05). In a previous study by Morone and co-workers [18], a 70% ethanolic extract by *Cyanobium* sp. LEGE 07175 exhibited 16.31 μg g^−1^ of dry biomass, with the best value being presented by *Synechocystis salina* LEGE 06099, with 49.82 μg g^−1^ of dry biomass. In this work, in terms of dry biomass, acetonic extracts of *Cyanobium* sp. LEGE 07175 presented a value of 799 μg g^−1^ and *Leptothoe* sp. LEGE 181156 a value of 498 μg g^−1^. Zeaxanthin’s properties include photoprotection in cyanobacteria and reduction of oxidative damage in humans’ eyes [19,20]. 

Myxoxanthophyll was detected in all cyanobacteria acetonic extracts and most aqueous extracts, except for *Cyanobium* sp. LEGE 07175. In contrast, echinenone was only detected in *Cyanobium* sp. LEGE 07175 with a concentration of 1.17 μg mg^−1^_dry extract_. This same carotenoid was detected in a 70% ethanol extract of *Cyanobium gracile* LEGE 12431 (7.31 μg mg^−1^_dry extract_) [21]. Echinenone was also found in other genera of cyanobacteria [18,21]. According to previous studies, this carotenoid is important in cyanobacteria photoprotection [19].

Beta-carotene is a terpene of relevant importance due to its important role as an antioxidant. This carotenoid was found in all strains in both acetone and aqueous extracts. For acetonic extracts, the concentrations ranged from 19.58 to 47.89 μg mg^−1^_dry extract_. There was a significant difference among the strain *Leptothoe* sp. LEGE 181156 and the Synechococcales, strains LEGE 181158, LEGE 181157, and LEGE 181150, which did not differ significantly from each other. These results may corroborate a study that analyzed the patterns of carotenoid synthesis in different genera and concluded that strains of the same genus present the same patterns of pigments production, namely concerning β-carotene [22]. Previous studies also reported β-carotene in different genera of cyanobacteria, namely, *Synechocystis, Nodosilinea, Phormidium, Leptolyngybya-like, Cuspidothrix, Anabaena, Nostoc, Aphanothece,* and *Gloeothece* [18,21,23,24,25], which validates the broad spectrum of cyanobacterial genera that produce this compound, and thus its importance in industries such as cosmetics. 

Morone et al., 2020 [18], showed that a 70% ethanol extract of *Cyanobium* sp. LEGE 07175 presented 8.06 μg of β-carotene per g of dry biomass. Accounting the extraction yield, in the present work, the acetonic extract of *Cyanobium* sp. LEGE 07175, which exhibited 19.58 μg mg^−1^_dry extract_, presented a value of 295 μg g^−1^ in terms of dry biomass. Another example is *Leptothoe* sp. LEGE 181156, which exhibited the highest value for β-carotene (47.89 μg mg^−1^_dry extract_) and, in terms of dry biomass, showed a value of 599 μg g^−1^, for acetonic extract. 

As β-carotene, γ-carotene was also detected in the acetonic extract of all species studied, although below the limit of quantification in *Cyanobium* sp. LEGE 07175. In addition to the antioxidant potential, β-carotene and γ-carotene are precursors of retinol, which works as a storage form of vitamin A and can be transformed into other modes of activation, such as retinoic acid and retinal [26]. In skin, retinoids are widely used in the treatment of diseases such as cancer, psoriasis, acne, ichthyosis, and even wrinkles due to their effect on cell differentiation, proliferation, and apoptosis [27]. Therefore, β-carotene and γ-carotene are two interesting carotenoids to be explored in cosmetics [28]. 

In this study, we found that zeaxanthin, β-carotene, and γ-carotene were the main carotenoids detected in most of the samples analyzed. A relevant aspect in the production of carotenoids in cyanobacteria is that by changing the culture conditions, it is possible to increase the production of these compounds. A previous study involving the optimization of culture conditions for cyanobacteria revealed that high light radiation may increase the production of carotenoids, such as β-carotene, echinenone, and myxoxanthophyll, and may exhibit strong protection under photo-oxidative conditions [29]. The results from the present study indicated that, despite the low yield achieved with the acetonic extraction (around 1.5 to 2% per gram of dry biomass (Table 7), the amount of carotenoids extracted with acetone may be profitable when compared to water, which presented percentages at least 20 times higher.

Amongst the pigments analyzed herein, the total chlorophylls content, including chlorophyll-*a* and its derivatives, was lower than total carotenoids, except for aqueous extracts. Unlike the results of our study, Lopes et al., 2020 [21], showed that chlorophylls were dominant over carotenoids in both, ethanol and acetonic extractions; however, the strains evaluated by the author were of terrestrial origin, which may explain the differences. Chlorophylls, as well as carotenoids, act as antioxidants and are widely used in the food, cosmetic, and pharmaceutical industries [30].

#### 3.1.2. Total Phenolic Content (TPC)

Despite the limitations inherent to colorimetric assays, the determination of TPC by the Folin–Ciocalteu method is widely used to determine total phenols in plant extracts, as well as to infer about their antioxidant potential. Overall, acetonic extracts had higher phenolic content than aqueous ones. Although phenolic compounds are soluble in solvents with higher polarity, the fact that the extraction was carried out sequentially with acetone and water led to the extraction of most of the compounds with acetone. The highest TPC was found in the acetone extract of *Leptothoe* sp. LEGE 181156, with 27.13 μg GAE mg^−1^ _dry extract_ with no significant differences when compared to Synechococcales cyanobacterium LEGE 181158 and *Cyanobium* sp. LEGE 07175 (24.71 and 23.11 μg GAE mg^−1^
_dry extract_, respectively). In a previous study by Morone et al. [18], using a 70% ethanol extract, *Cyanobium* sp. LEGE 07175 had a value of 1.09 mg GAE g^−1^_dry biomass_, which was higher than the result obtained herein for the same strain (0.29 mg GAE g^−1^, converted according to extraction yield). Another study with the same genus [31], revealed values around 82 μg GAE mg^−1^_dry extract_ for acetonic extracts of another *Cyanobium* sp. Favas et al. [14], presented a value of 17.59 μg GAE mg^−1^_dry extract_ for acetonic extract of *Leptolyngbya* cf. *ectocarpi* LEGE 11479. 

Regarding aqueous extracts, although the values are lower, between 14.49 to 15.67 μg GAE mg^−1^_dry extract_, when converted to dry biomass, the results turn out to be promising, considering the excellent yield that aqueous extraction may provide; for example, this was true for Synechococcales *cyanobacterium* LEGE 181150 and Synechococcales cyanobacterium LEGE 181157, which were the strains that presented the best results (6.48 and 4.33 mg GAE g^−1^_dry biomass_, respectively). However, regarding acetonic extracts, the results were much lower for the same strains in terms of dry biomass (0.39 and 0.30 mg GAE g^−1^_dry biomass_, respectively). Phenolic compounds are known for their beneficial effect against humans, including skin diseases, namely due to their anti-inflammatory and antioxidant potential [32]. In the skin, phenolic compounds were found to be active as antioxidants and anti-inflammatories [33] and are effective at inhibiting the enzyme tyrosinase, and thus at treating aesthetic problems such as hyperpigmentation [34]. Regarding marine cyanobacteria, as far as we are aware, there are no other previous reports on the TPC, corroborating the importance of further exploitation of cyanobacteria extracts concerning antioxidant activity.

#### 3.1.3. Proteins

Proteins play a key role in the structure and integrity of the skin. The main benefit of using proteins in cosmeceuticals is related to improving skin hydration, as they act as water-binding molecules. In addition to being responsible for forming collagen, which is a protein that gives firmness to skin and prevents the formation and exacerbation of wrinkles and expression lines, proteins are the main basis for the formation of new tissues, also being implicated in improving wound healing. Furthermore, proteins present UV protection capacity, antimicrobial, and antioxidant properties. Therefore, there is a growing search for products with antiaging action based on bioactive peptides [35]. In this study, the characterization of the total protein content of the extracts studied was carried out. It was confirmed that the aqueous extracts presented much higher values than the acetonic ones, with emphasis on strains used in this work within the order Synechococcales, which showed promising results. Synechococcales cyanobacterium LEGE 181157 ranked first with 760.81 μg (BSA) mg^−1^_dry extract_ (*p* < 0.05), followed by Synechococcales cyanobacterium LEGE 181150 (*p* < 0.05) and Synechococcales cyanobacterium LEGE 181158 (695.25 and 579.99 μg (BSA) mg^−1^_dry extract_, respectively). Converting to extraction yield, the values are 220, 331, and 100 mg (BSA) g^−1^_dry biomass_, respectively. Therefore, in terms of biomass, Synechococcales cyanobacterium LEGE 181150 exhibited the highest value. Favas and coworkers, 2022 [14] also reported higher values for aqueous extracts than acetonic ones, in which the freshwater strain *Cephlothrix lacustris* LEGE 15493 presented 521.18 μg (BSA) mg^−1^_dry extract_.

#### 3.1.4. Phycobiliproteins (PBPs)

Phycobiliproteins are water-soluble proteins, which are deep-colored, formed by a complex between proteins and covalently bound phycobilins. Phycocyanin (PC), allophycocyanin (APC), and phycoerythrin (PE) are the three main PBPs groups, absorbing light within specific regions of the spectrum, according to their structure and types of bilins [9]. In recent years, the demand for microorganisms to synthetize natural colorants for cosmetics has increased in order to replace synthetic ones used in lipsticks, eyeliners, and blushes [2]. The PBPs PC, of blue color; APC, blue-green; and PE and PEC, both of red/pink color, have been already used as a natural coloring in foods, nutritional supplements, cosmetics, and as fluorescent markers in immunoassays [36].

As water-soluble compounds, PBPs were only analyzed in aqueous extracts. As visually confirmed through its intense pink color, the aqueous extract of the strain Synechococcales cyanobacterium LEGE 181157 was the richest in PE (275.04 μg mg^−1^ _dry extract_) followed by the strain *Leptothoe* sp. LEGE 181156 (159.05 μg mg^−1^ _dry extract_) (*p* < 0.05). Validated by the strong blue color, Synechococcales cyanobacterium LEGE 181150 aqueous extract was the richest in PC (222.76 μg mg^−1^ _dry extract_) (*p* < 0.05). Regarding APC, Oscillatoriales cyanobacterium LEGE 181159 presented the highest value, followed by Synechococcales cyanobacterium LEGE 181150 (65.90 and 57.29 μg mg^−1^ _dry extract_, respectively). Only *Cyanobium* sp. LEGE 07175, with a light green extract, showed significantly lower values. In terms of dry biomass, Synechococcales cyanobacterium LEGE 181157 exhibited 113.18 mg g^−1^ dry biomass for PE and Synechococcales cyanobacterium LEGE 181150 83.53 mg g^−1^ _dry biomass_ for PC. These values are in accordance with literature data, namely for PE concentration in the thermotolerant cyanobacterium *Leptolyngbya* sp. KC45 (100 mg g^−1^) [37]. For PC, the value described is within the concentrations obtained with different strains of Spirulina, as exposed in a recent review by Jaeschke and co-workers [38] in which different methods of PC extraction from Spirulina strains are compared. Concerning total PBPs by dry biomass, the results obtained are also in accordance with reported data for the strain Synechococcales cyanobacterium LEGE 181157 (179 mg g^−1^_dry biomass_) compared with in the strain *Anabaena* sp. NCCU-9 (91 mg g^−1^_dry biomass_) [39] and *Cyanobium* sp. (200 mg g^−1^_dry biomass_) [31]. The results obtained confirm the use of cyanobacteria to obtain natural dyes and expand the range of genera with this potential. Along with the coloring role, PBPs are described as potent antioxidant and anti-inflammatory agents, properties that are beneficial in cosmetic formulations [40]. 

#### 3.1.5. Antioxidant Potential

In this work, the relevant physiological free radical O_2_^•−^, one of the major ROS that provoke oxidative damage in the human body, was used to determine the antioxidant potential of cyanobacteria extracts. In terms of radical scavenging, the aqueous extract of the strain Synechococcales cyanobacterium LEGE 181157 was the most effective, presenting the best IC_50_ value of 63.24 μg mL^−1^_dry extract_, followed by Synechococcales cyanobacterium LEGE 181150 (112.18 μg mL^−1^_dry extract_) (*p* > 0.05). Synechococcales cyanobacterium LEGE 181157 was also the strain that presented the highest value for total proteins (760.81 μg BSA mg^−1^_dry extract_) (*p* < 0.05), followed by Synechococcales cyanobacterium LEGE 181150, with 695.25 μg BSA mg^−1^_dry extract_. Moreover, Synechococcales cyanobacterium LEGE 181157 showed the highest value for PE (275.04 μg mg^−1^_dry extract_) (*p* < 0.05), and Synechococcales cyanobacterium LEGE 181150 presented the highest content for PC (222.76 μg mg^−1^_dry extract_) (*p* < 0.05). According to the statistical analyses, a significant negative correlation was found between the total proteins and the IC values (−0.982, *p* < 0.01 for IC_50_ and −0.739, *p* < 0.01 for IC_25_). In addition, a negative correlation was also observed between PE and IC values, and the same occurred for the TPC. Furthermore, a significant negative correlation was found between ICs and PC (−0.680, *p* < 0.05 and −0.749, *p* < 0.01, for IC_50_ and IC_25_, respectively). Although *Leptothoe* sp. LEGE 181156 was the only strain that did not reach the IC_50_ for aqueous extract, this strain managed to obtain more than 40% of enzymatic inhibition at the concentration 104.00 μg mL^−1^_dry extract_, and its IC_25_ reached 41.10 μg mL^−1^_extract dry_ (Figure 2). This antioxidant activity observed in the aqueous extracts may be attributed to their richness in proteins and PBP.

Regarding the acetonic extracts, strains belonging to Synechococcales cyanobacterium order, followed by *Leptothoe* sp. LEGE 181156, showed the best results for free-radical sequestration (*p* > 0.05). However, the values were much higher than those displayed by the aqueous extract. These acetonic extracts are richer in carotenoids than aqueous extracts (Table 1). According to the statistical analyses, although they lacked statistical significance, negative correlations were noted with total carotenoids content, lutein, β-carotene, and γ-carotene. Favas et al. [14], using both extractions, also reported a higher effectiveness for the aqueous extracts when compared to the acetonic ones. In their work, aqueous extracts of *Cephalothrix lacustris* LEGE 15493 showed an IC_50_ of 65.50 μg mL^−1^_dry extract_, while the IC_50_ of its acetonic extract was not detected. Morone et al. [18], who studied ethanol extracts of other species, reported the lowest IC_50_ value for *Phormidium* sp. LEGE 05292 (822.70 μg mL^−1^). Amaro and co-workers [23] revealed that *Scenedesmus obliquus* (M2-1) showed an IC_50_ of 826 μg mL^−1^ for acetonic extract. According to the studies published so far, aqueous extracts have shown great potential in terms of their ability to scavenge O_2_^•−^. In this way, we may consider the interest of the aqueous extracts of the strains explored in our work for cosmeceutical applications.

#### 3.1.6. Inhibition of Metalloproteinases (MMPs)

The dermis ECM is extremely important for the maintenance of skin structure. When collagen, elastin, and HA are degraded or deficient, the effects are expressed in the visual appearance of the skin. Under oxidative stress, an inflammatory response with activation of MMPs such as collagenase, elastase, and HAase is triggered, contributing to the degradation of these ECM vital components, and promoting wrinkles formation [1,41]. Tyrosinase is another enzyme that is also activated mainly by external factors, such as exposure to UV rays. This enzyme acts in photoprotection as a catalyst in the synthesis of melanin, but when overactivated, it causes pigmentation lesions, such as vitiligo, or hyperpigmentation marks, such as melasma and age spots [42]. Triggered by the importance of MMPs inhibitors in skin, the ability of the aqueous and acetonic extracts to inhibit these key enzymes was explored.

Regarding HAase, the lowest results for IC_50_ were presented by the aqueous extracts of the species belonging to the order Synechococcales, in which the strain Synechococcales cyanobacterium LEGE 181150 presented the best value (483.86 μg mL^−1^_dry extract_). For acetonic extracts, the results were around 750 μg mL^−1^_dry extract_. Regarding statistical analysis, there is a strong negative correlation between total proteins and the IC_50_ for both aqueous and acetonic extracts (−0.593, *p* < 0.05 and −1.000 *p* < 0.01, respectively) and a negative correlation between TPC and IC_50_, also for both extracts, suggesting that both proteins and phenols may be implicated in the inhibition of this enzyme. Synechococcales cyanobacterium LEGE 181150 was the strain exhibiting one of the highest total proteins for both extractions, which corroborates the deductions above. In this study, the value of total proteins for aqueous extracts of *Cyanobium* sp. was not as high as that presented by the strains belonging to Synechococcales order, which is also in line with the statistical correlations found. Our results highlight the importance of aqueous extracts as potential antiaging ingredients, possibly due to their protein content. Although the aqueous extract of *Cyanobium* sp. LEGE 07175 displayed an IC_50_ of 894.59 μg mL^−1^, in a previous study realized by us [18], a 70% ethanol extract of this strain, *Cyanobium* sp. LEGE 07175, showed stronger inhibitory activity (IC_50_ = 208.36 μg mL^−1^). This may be due to a synergistic effect between the compounds extracted by ethanol 70%, since this solvent has more affinity to a mixture of polar and less polar compounds, contrary to water, from which compounds of more polar nature are mainly extracted. Furthermore, Pagels et al. [43], showed considerable IC_50_ for aqueous and acetonic extracts, using another strain of the genus *Cyanobium* (67.25 and 108.74 μg mL^−1^, respectively). Favas et al. [14], presented the lowest value for aqueous extract of *Leptolyngbya* cf. *ectocarpi* LEGE 11479 (863 μg mL^−1^_dry extract_) and only IC_25_ for acetonic extracts. A previous work focusing polysaccharides from cyanobacteria [13] reported that *Nostochopsis lobatus* MAC0804NAN had a notable IC_50_ of 7.18 μg mL^−1^ for HAase inhibition. These results caught our interest since, considering the polar nature of polysaccharides, these compounds may also be present in the aqueous extracts and, together with proteins, contribute to the inhibition of the enzyme. 

Inhibition of elastase was only achieved with acetonic extracts, Synechococcales cyanobacterium LEGE 181157 being the unique one that reached IC_50_ value (380.50 μg mL^−1^_dry extract_. In addition, its IC_25_ presented an interesting value of 51.43 μg mL^−1^_dry extract_. On the other hand, this strain was not able to inhibit collagenase. For acetonic extracts, collagenase inhibition was achieved at 750 μg mL^−1^ for *Leptothoe* sp. LEGE 181156 and *Cyanobium* sp. LEGE 07175 (36.61 and 32.88%, respectively), and at 1000 μg mL^−1^ for Synechococcales cyanobacterium LEGE 181150, with 44.40%. Favas et al. [14], reported that acetonic extracts of *Leptolyngybya* cf. *ectocarpi* LEGE 11479 reached IC_50_ (391 μg mL^−1^_dry extract_) for elastase inhibition, and Pagels et al. [43] showed collagenase inhibition for aqueous extract of the strain *Cyanobium* sp. LEGE 06113 (IC_50_ = 582.82 μg mL^−1^) but with no results below 1000 μg mL^−1^ for elastase and tyrosinase inhibition. Regarding aqueous extracts, Synechococcales cyanobacterium LEGE 181158 presented 25.97% of inhibition at 1000 μg mL^−1^. As presented above, this strain also exhibited inhibition of hyaluronidase.

Finally, IC_50_ for tyrosinase was displayed for the acetonic extracts for two strains. *Leptothoe* sp. LEGE 181156 was the only strain that exhibited IC results for both tyrosinase (IC_50_ = 465.92 μg mL^−1^ and IC_25_ =133.47 μg mL^−1^) and elastase (IC_25_ = 378.26 μg mL^−1^). In addition, this strain also showed collagenase inhibition of 36.61% at a concentration of 750 μg mL^−1^_dry extract_. With respect to Synechococcales cyanobacterium LEGE 181157, ICs values for hyaluronidase, elastase and tyrosinase inhibition were found, the IC_25_ being more attractive (613.47, 51.43, and 381.33 μg mL^−1^_dry extract_, respectively). A more detailed analysis showed a significant negative correlation between TPC (−0.972, *p* < 0.05), Total Proteins (−0.984, *p* < 0.05) and Total Carotenoids (−0.974, *p* < 0.05), and IC_50_. Additionally, zeaxanthin, β-carotene (−0.956, *p* < 0.05), and γ-carotene showed a negative correlation in relation to tyrosinase inhibition. For the same enzyme, Favas et al. [14], reported that only the acetonic extract of *Nodosilinea nodulosa* LEGE 06104 was able to reach IC_50_ (989.26 μg mL^−1^_dry extract_). Ethanol extracts studied by Morone et al. [18] did not show tyrosinase inhibition.

In a general way, extracts richer in proteins seem to be more effective for free radicals scavenging, being more attractive to counteract oxidative stress, while acetonic extracts, which are richer in carotenoids and phenols, seem more effective as enzyme inhibitors. Regarding phenolic compounds, it seems that they contribute to both biological activities evaluated.

## 4. Materials and Methods

### 4.1. Cyanobacteria Strains and Biomass Production

Seven cyanobacterial strains isolated from the Portuguese and Cape Verde marine ecosystems and maintained in the Blue Biotechnology and Ecotoxicology Culture Collection (LEGE-CC) at the Interdisciplinary Centre of Marine and Environmental Research (CIIMAR) [44] were randomly chosen for this study, with the aim of conducting a more comprehensive screening of the most promising cyanobacteria strains. The strains panel included the picoplanktonic *Cyanobium* sp. LEGE 07175 from Portuguese marine ecosystem and the filamentous Synechococcales cyanobacterium LEGE 181150 [45], *Leptothoe* sp. LEGE 181155, *Leptothoe* sp. LEGE 181156, Synechococcales cyanobacterium LEGE 181157, Synechococcales cyanobacterium LEGE 181158, and Oscillatoriales cyanobacterium LEGE 181159, from Cape Verde marine ecosystems, which are in the process of publication. For biomass production, a scale-up culture scheme up to 4 L was set. The strains were grown in Z8 medium [46], supplemented with 10 µg/L vitamin B12 and 25 g/L of synthetic sea salts (Tropic marin, Berlin, Germany). Cultures were maintained at 25 °C, with a light intensity of 10–30 μmol photons m^−2^ s^−1^ and with a photoperiod of 14h light:10h dark. The fresh biomass was collected by filtration, frozen, freeze-dried, and stored at −20 °C until extracts preparation. 

### 4.2. Extract Preparation

Acetone and aqueous extracts were sequentially prepared from each strain by following the procedure described by Favas et al. [14]. First, the acetonic extract was prepared using 2 g of dry biomass. The biomass was suspended in 80 mL of acetone and extracted for 10 min in an ultrasonic bath (Fisherbrand^®^-FB15053, Loughborough, UK). The supernatant was collected, and the biomass was reextracted three more times. After the acetonic extraction, the resulting pellet was left to dry in the fume hood until the remaining solvent was completely evaporated. The pellet was further extracted with 70 mL of distilled water, following the same procedure. Cell debris was removed by centrifugation (5000 Gs, 5 min, 4 °C) (Thermo ScientificTM HERAUS MegafugeTM 16R, Waltham, MA, USA). Supernatants from each extraction were evaporated under reduced pressure (acetone) (BUCHI R-210 Rotary Evaporator, Cambridge, MA, USA) or frozen and lyophilized (water). The dry extracts were kept at −20 °C until further chemical and biological analysis. The extractions yield is displayed below (Table 7).

### 4.3. Phytochemical Analysis

#### 4.3.1. Determination of Pigments Profile by HPLC-PDA

Dried cyanobacteria extracts were dissolved in HPLC-grade methanol (acetonic extract) or Milli-Q water (aqueous extracts) to a final concentration of 5 and 10 mg mL^−1^, respectively, and filtered through a 0.2 μm pore membrane. Carotenoids analysis was performed following the method previously described [47], with slight modifications. A Waters Alliance 2695 high-performance liquid chromatography (HPLC) with photodiode array (PDA) detector (USA) was employed to resolve, detect, and identify the compounds of interest. The stationary phase was a YMC Carotenoid C30 (250 × 4.6 mm; 5 μm) column, kept at constant temperature (25 °C) with a column heater (Waters Corporation, Milford, CT, USA). The mobile phase consisted of 2 solvents: methanol (A) and tert butyl methyl ether (B) (VWR Prolabo), starting with 95% A and installing a gradient to obtain 10% B at 5 min, 18% B at 20 min, 30% B at 28 min, 50% B from 31 to 37 min, 80% B from 38 to 47 min, and 5% B from 48 to 50 min. The flow rate was 0.90 mLmin^−1^ and the injection volume was 5 μL. Data were processed using Empower chromatography software (Waters, USA). Spectra data from all peaks were collected in the range 250 to 750 nm. 

Compounds were identified by comparing their retention times and UV-Vis spectra with those of authentic standards. Carotenoids quantification was achieved by measuring the absorbance recorded in the chromatograms relative to external standards at 450 nm. 

Zeaxanthin, lutein, echinenone, myxoxanthophyll, phaeophytin-*a*, β-carotene, γ-carotene, and chlorophyll-*a* (Extrasynthese, Genay, France; Sigma-Aldrich, St. Louise, MO, USA; DHI, Horsholm, Denmark) were quantified with the authentic standards; unidentified carotenoids were quantified as zeaxanthin, the most representative xanthophyll, and chlorophyll derivatives and phaeophytin-*a* as chlorophyll-*a*, the major cyanobacteria chlorophyll. Calibration curves were performed with five different concentrations of standards, selected as representative of the range of compounds concentrations in the samples. The calibration plots and r^2^ values for the analyzed carotenoids and chlorophyll-*a* are shown in Table 8.

#### 4.3.2. Total Phenolic Content (TPC)

The TPC of the cyanobacterial extracts was determined using the colorimetric assay of Folin–Ciocalteu, according to Barroso et al. [18,48]. The acetonic extracts were solubilized in DMSO and the aqueous extracts in water. Briefly, a volume of 25 μL of each extract (10 mg mL^−1^) was thoroughly mixed with 25 μL of Folin–Ciocalteu reagent (Sigma-Aldrich, St. Louis, MO, USA), 200 μL of Na_2_CO_3_ solution (75 g L^−1^) and 500 μL of deionized water. After the incubation period (60 min at room temperature), the absorbance of the colored product was measured at 725 nm, using a Synergy HT Multi-detection microplate reader (Biotek, Bad Friedrichshall, Germany) operated by GEN5^TM^ software. Standard calibration curves (*y* = 2.097*x* + 0.01560, R^2^ = 0.9989, for aqueous extracts and *y* = 2.204*x* + 0.01401, R^2^ = 0.9982, for acetonic extracts) were obtained with seven concentrations of gallic acid (GA) (0.025 to 0.5 mg mL^−1^). TPC in each extract was expressed in µg of gallic acid equivalents (GAE) per mg of dry biomass. Three independent determinations were carried out in triplicate.

#### 4.3.3. Total Proteins

Total proteins concentration was determined using the BSA Protein Assay kit (n° 23227, Thermo-Scientific, Waltham, MA, USA) by following the manufacturer’s instructions and according to Favas et al. [14]. Aqueous extracts were prepared in water, while acetone extracts were prepared in DMSO. Briefly, in a 96-well plate, 25µL of each extract (1 mg mL^−1^) was mixed with 200 µL of working reagent. The absorbance was measured at 562 nm using a Synergy HT Multi-detection microplate reader (Biotek, Bad Friedrichshall, Germany) operated by GEN5^TM^ software. Standard curves (*y* = –126.87*x*^3^ + 547.73*x*^2^ + 483.85*x* − 10.017; R^2^ = 0.999 and *y* = 162.87*x*^3^ − 248.51*x*^2^ + 932.13*x* − 11.715; R^2^ = 0.999) were obtained for each extract (aqueous and acetonic, respectively), using nine concentrations of albumin (BSA) (25 to 2000 µg mL^−1^) to quantify the proteins. Three independent experiments were carried out in triplicate. The total proteins were expressed as µg of bovine serum albumin (BSA) equivalents per mg of dry extracts.

#### 4.3.4. Phycobiliproteins

The pigments present in aqueous extracts were determined spectrophotometrically. Aqueous extracts were resuspended in water. PBPs were determined by measuring the absorbances at different wavelengths (562, 615 and 645 nm), in a cell with 1 cm of optical path. The corresponding formulas were applied, as previously described by Pagels et al. [31]:$$Phycocyanin\left(PC\right)=\frac{A_{615nm}-0.474\times A_{652nm}}{\begin{array}{c} 5.34 \end{array}}\;Allophycocyanin\left(APC\right)=\frac{A_{652nm}-0.208\times A_{615nm}}{\begin{array}{c} 5.09 \end{array}}\;Phycoerythrin\left(PE\right)=\frac{A_{562nm}-2.41\times PC-0.849\times APC}{\begin{array}{c} 9.62 \end{array}}$$

Aqueous extracts were resuspended to a final concentration of 0.5 mg mL^−1^. The experiment was carried out in triplicate, and the results were expressed in µg of the respective phycobiliprotein per mg of dry extract.

### 4.4. Cell Assays

#### 4.4.1. Cell Culture

To assess the safety of the cyanobacteria acetonic and aqueous extracts, a preliminary in vitro cytotoxicity assay was performed using three different cell lines, the human keratinocytes HaCAT (ATCC), the mice fibroblasts 3T3L1 (ATCC) and the human endothelial cells hCMEC/D3 (provided by Dr. PO Couraud (INSERM, Paris, France). Cell culture was performed as already described by us in Morone et al. and Favas et al. Briefly, cells were cultured in DMEM Glutamax medium (Dulbecco’s Modified Eagle Medium DMEM GlutaMAXTM—Gibco, Glasgow, UK), supplemented with 10% (*v/v*) of fetal bovine serum (Biochrom, Berlin, Germany), 0.1% of amphotericin B (GE Healthcare, Little Chafont, UK) and 1% of Pen-Strep (penicillin-streptomycin, 100 IU mL^−1^ and 10 mg mL^−1^, respectively) (Gibco, Berlin, Germany). Cells maintenance and assays were performed in a humidified atmosphere of 5% CO_2_ at 37 °C. Between 80–90% cell confluence, adherent cells were washed with phosphate-buffered saline (PBS, Gibco), detached with a small amount of TrypLE Express enzyme (Gibco), passed for maintenance, and seeded for the planned assays.

#### 4.4.2. Cytotoxicity—MTT Assay

Cytotoxicity assay was performed by measuring the reduction of 3-(4,5-dimethylthiazole-2-yl)-2,5-diphenyltetrazolium bromide (MTT), as previously described [18]. Keratinocytes, fibroblasts and endothelial cells were seeded in 96-well plates at a density of 2.5 × 10^4^ cells mL^−1^; 3.3 × 10^4^ cells mL^−1^ and 1.0 × 10^5^ cells mL^−1^, respectively. After 24 h of adhesion, the culture medium was removed and cells were exposed for 24 and 48 h to fresh medium supplemented with 1% of extracts in five serial concentrations, from 12.5 to 200 μg mL^−1^. Acetone extracts were prepared in dimethyl sulfoxide (DMSO, Gibco) and diluted with DMEM prior to cells exposure; the maximum DMSO concentration did not exceed 1%. DMSO at 1% and 20% were used as solvent and positive controls, respectively. Aqueous extracts were prepared in PBS and diluted with DMEM prior to cells exposition. PBS at 1% and DMSO at 20% were used as solvent and positive controls, respectively. After each incubation time, 20 μL of 1 mg mL^−1^ MTT (Sigma-Aldrich), was added to each well and incubated at 37 °C for 3h. Following incubation, the media were removed, and the purple-colored formazan salts were dissolved in DMSO. The absorbance was read at 550 nm in a Synergy HT Multi-detection microplate reader (Biotek, Bad Friedrichshall, Germany) operated by GEN5^TM^ software. The assay was run in quadruplicate and averaged. Cytotoxicity was expressed as a percentage of cell viability, considering 100% viability in the solvent control. For reproducibility, each assay was independently repeated three times.

### 4.5. Biological Activities

#### 4.5.1. Superoxide Anion Radical (O_2_^•−^) Scavenging

The superoxide anion radical (O_2_^•−^) scavenging activity of the extracts was determined as described previously [49] with some modifications. The acetone extracts were resuspended in DMSO, while the aqueous extracts were resuspended in water. Six serial dilutions, from 10 mg mL^−1^, were prepared for each extract and tested in order to evaluate the extracts’ behavior and IC values. All reagents were dissolved in phosphate buffer (19 µM, pH 7.4). A volume of 50 µL of serial dilutions of the cyanobacteria extracts (0.104 to 1.667 mg mL^−1^ for acetonic extracts; and 0.013 to 1.667 mg mL^−1^, for aqueous extracts) was mixed with 50 µL of 166 µM β-nicotinamide adenine dinucleotide reduced-form (NADH) solution and 150 µL of 43 µM nitrotetrazolium blue chloride (NBT), in a 96-well plate. A volume of 50 µL of 2.7 µM phenazine methosulphate (PMS) was added to each well. The radical scavenging activity of the samples was monitored with a Synergy HT Multi-detection Microplate Reader operated by GEN5^TM^ (Biotek, Bad Friedrichshall, Germany), in kinetic function, at room temperature, for 2 min, at 562 nm. All reagents were dissolved in phosphate buffer (19 μM, pH 7.4). Three independent assays were performed in triplicate. GA was used as positive control. The results were expressed as a percentage of radical scavenging in comparison to the untreated control. The results for the calculated IC values were expressed as mean ± SD (µg/mL) of at least three independent assays performed in duplicate. The IC values and the corresponding dose–response curves were calculated with Graphpad Prism^®^ software (version 9, for MacOS).

#### 4.5.2. Hyaluronidase Inhibition

HAase Inhibition assay was determined as proposed by Ferreres et al., [50] with some modifications. The acetone extracts were resuspended in DMSO, while the aqueous extracts were resuspended in water. Six serial dilutions were prepared for each extract (0.125 to 1.0 mg mL^−1^ and 0.063 to 1.0 mg mL^−1^, respectively) and tested to evaluate the extracts’ behavior and IC values. Twenty-five microliters of each sample were added to each reaction tube. A volume of 175 µL of HA solution (0.7 mg mL^−1^ in water:buffer, 5:2 *v*/*v*, kept at 37 °C) was added to each reaction tube and gently mixed. The reaction started by adding 25 μL of HAase (900 U mL^−1^ in NaCl 0.9%). After 30 min incubation at 37 °C, the enzymatic reaction was stopped with 25 μL of disodium tetraborate 0.8 M, followed by subsequent heating for 3 min in a boiling water bath. After cooling to room temperature, 375 µL of DMBA solution was added and gently mixed (2 g of DMAB dissolved in a mixture of 2.5 mL of 10 N HCl and 17.5 mL of glacial acetic acid and further diluted 1:2 with glacial acetic acid immediately before use). The tubes were incubated at 37 °C for 20 min and the absorbance of the colored product was measured at 560 nm in a Synergy HT Multi-detection microplate reader (Biotek, Bad Friedrichshall, Germany) operated by GEN5^TM^ software. Negative control was performed in the absence of extract. Disodium cromoglycate (DSCG) was used as positive control. Three independent assays were performed in triplicate.

#### 4.5.3. Elastase Inhibition

Porcine pancreatic elastase inhibition assay was slightly modified from those proposed by Mota et al. [51]. Acetonic and aqueous extracts were resuspended in DMSO and water, respectively. Briefly, in a 96-well plate, 50 µL of extract (0.05 to 0.4 mg mL^−1^ for each extract) was mixed with 90 µL of HEPES buffer (0.1 M), 10 µL of N-succinyl-Ala-Ala-Ala *p*-nitroanilide substrate (100 µM), 70 µL of acetate buffer (200 mM), and 30 µL of elastase (1 U mL^−1^). The plate was incubated at 37 °C for 10 min, and the absorbance of the reaction product was measured at 405 nm, in a Synergy HT Multi-detection microplate reader (Biotek, Bad Friedrichshall) operated by GEN5^TM^. Negative control was performed in the absence of extract, and ascorbic acid was used as positive control. Three independent assays were performed in triplicate. The results were expressed as a percentage of enzyme inhibition in comparison to the untreated control.

#### 4.5.4. Collagenase Inhibition

Collagenase inhibition assay was determined as reported by Andrade et al. [52] with some modifications. The substract *N*-(3-furyl-acryloyl)-Leu-Gly-Pro-Ala (FALGPA) 1 mM was dissolved in tricine buffer (50 mM, pH 7.5). Collagenase enzyme was prepared in the buffer at 1 U mL^−1^. First, 10 μL of extract dilutions (0.75 and 1.0 mg mL^−1^), 45 μL of tricine buffer, and 50 μL of collagenase (1 U mL^−1^) were added to a 96-well plate and kept at 37 °C for 15 min. The reaction was started by adding 120 μL of FALGPA. The substrate hydrolysis was monitored with a Synergy HT Multi-detection Microplate Reader operated by GEN5^TM^ (Biotek, Bad Friedrichshall, Germany), operating in kinetic function, at room temperature, for 8 min, at 345 nm. Negative control was performed in the absence of extract, and gallic acid was used as positive control. Results were expressed as percentage of enzyme inhibition in comparison to the untreated control. The assay was performed in triplicate. 

#### 4.5.5. Tyrosinase Inhibition

The tyrosinase inhibition assay was performed according to Adhikari and co-workers [53], with some modifications. Five serial dilutions, from 0.063 to 1.0 mg mL^−1^, were prepared for each extract. Briefly, in a 96-well plate, 20 µL of each extract was mixed with 100 µL of tyrosinase (30 U mL^−1^ in phosphate buffer). Acetonic extracts and aqueous extracts were resuspended in DMSO and water, respectively. The mixture was incubated at 30 °C during 8 min. After that, 80 µL of L-DOPA (L-3,4-dihydroxyphenylalanine) solution (2.5 mM in phosphate buffer) was added, and the absorption was immediately read (T0) with a Synergy HT Multi-detection microplate reader (Biotek, Bad Friedrichshall, Germany) operated by GEN5^TM^ software, at 475 nm. After 8 min, the absorbance was measured again (T8). Negative control was performed in the absence of extract, and kojic acid was used as positive control. Three independent assays were performed in triplicate. The results were expressed as a percentage of enzyme inhibition in comparison to the untreated control. 

### 4.6. Statistical Analysis

Statistical analysis was performed using IBM SPSS STATISTICS software, version 28.0.1.0, IBM Corporation, New York, NY, USA (2021). Data were analyzed for normality and homogeneity of variances using Kolmogorov–Smirnov and Leven’s tests, and then submitted to one-way ANOVA, using a Tukey’s HSD (honest significant difference) as a post hoc test. A Pearson correlation test was used to compare normalized expression data between the chemical profile and the biological activities of cyanobacteria extracts.

## 5. Conclusions 

Many studies have shown that, to protect themselves from the damage associated with extreme environmental conditions, such as high desiccation and UV radiation, cyanobacteria produce compounds such as carotenoids, phenolic compounds and phycobiliproteins.

This study revealed that the cyanobacteria strains evaluated stood out regarding cosmetic purposes and sustainability. The extractions were performed sequentially, allowing the obtention of different bioactive extracts from the same biomass. This increases biomass rentability, has a lower environmental impact and becomes economically attractive. The aqueous extracts of Synechococcales cyanobacterium LEGE 181157 and Synechococcales cyanobacterium LEGE 181150 strains, rich in phycobiliproteins, showed antioxidant activity, reaching an attractive IC_50_ for superoxide radical scavenging. Furthermore, these extracts had the ability to inhibit the enzyme HAase. As such, they seem interesting for antioxidant purposes. Moreover, PBPs, the major components of aqueous extracts, are photoprotective compounds, and their natural and attractive colors may add value in cosmetic products. On the other hand, phenols and carotenoids present in the acetonic extracts, especially in *Leptothoe* sp. LEGE 181156 and Synechococcales cyanobacterium LEGE 181150, demonstrated potential to preserve dermal matrix components through the inhibition of MMPs. Overall, the results obtained drive the studies on cyanobacteria towards the cosmetic industry, as a promising source of natural extracts with potential in cosmetic formulations.

## Figures and Tables

**Figure 1 marinedrugs-20-00761-f001:**
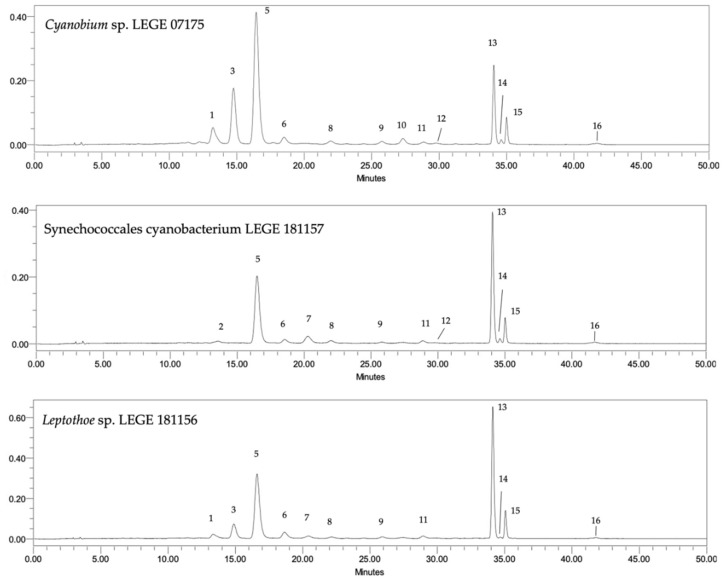
Carotenoid and chlorophyll profile of acetonic extracts of cyanobacteria strains. HPLC-PDA recorded at 450 nm. Unidentified carotenoids (**1**,**3**,**8**,**9**,**11**,**12**,**15**), Zeaxanthin (**5**), Chlorophyll-*a* derivative (**6**), Myxoxanthophyll (**7**), Echinenone (**10**), β-carotene (**13**), Phaeophytin-*a* (**14**), and γ-carotene (**16**).

**Figure 2 marinedrugs-20-00761-f002:**
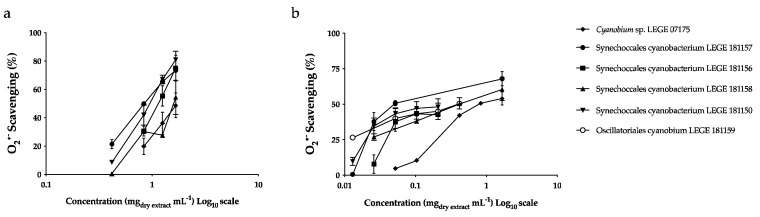
Superoxide anion radical (O_2_^•−^) scavenging activity of cyanobacteria acetonic (**a**) and aqueous (**b**) extracts. Values are expressed as the mean ± SD of at least three independent experiments, performed in duplicate.

**Table 1 marinedrugs-20-00761-t001:** Carotenoid and chlorophyll contents (μg mg^−1^ dry extract) in the aqueous and acetonic extracts of the cyanobacteria strains, determined by HPLC-PDA ^1,2^.

Peak	Compound	RT (min)	*Cyanobium* sp. LEGE 07175		Synechococcales cyanobacterium LEGE 181150		*Leptothoe* sp. LEGE 181156		Synechococcales cyanobacterium LEGE 181157		Synechococcales cyanobacterium LEGE 181158		Oscillatoriales cyanobacterium LEGE 191159
			Aqueous Extract	Acetonic Extract	Aqueous Extract	Acetonic Extract	Aqueous Extract	Acetonic Extract	Aqueous Extract	Acetonic Extract	Aqueous Extract	Acetonic Extract	Aqueous Extract
1	Unidentified carotenoid	13.35	nq	7.28 ± 0.03 ^a^	nd	2.55 ± 0.04 ^c^	nd	3.31 ± 0.07 ^b^	nd	nd	nd	nd	nd
2	Lutein	13.44	nd	nd	nd	nd	nd	nd	nd	2.07 ± 0.70	nd	1.96 ± 0.15	nd
3	Unidentified carotenoid	14.81	0.53 ± 0.05	21.03 ± 0.40 ^a^	0.04 ± 0.02	7.75 ± 0.39 ^b^	nq	8.53 ± 0.03 ^b^	nd	nd	nd	0.77 ± 0.12 ^c^	nd
4	Chlorophyll−*a*	17.02	nd	nd	14.77 ± 1.21 ^A^	32.19 ± 1.02	0.81 ± 0.06 ^C^	nd	0.06 ± 0.002	nd	9.42 ± 1.01 ^B^	31.96 ± 2.04	8.69 ± 0.07 ^B^
5	Zeaxanthin	17.15	1.58 ± 0.08	53.08 ± 1.48 ^a^	nd		nd	39.85 ± 2.69 ^b^		24.38 ± 0.33 ^c^	nd		nd
6	Chlorophyll−*a* derivative	17.98	nd	25.27 ± 1.17 ^b^	nd	29.23 ± 0.69 ^a,b^	nd	34.81 ± 1.52 ^a^	nd	12.69 ± 2.04 ^c^	nq	21.90 ± 3.55 ^b^	nd
7	Myxoxanthophyll	21.07	nd	nd	0.29 ± 0.03 ^A^	8.22 ± 0.37 ^a^	nd	4.07 ± 0.38 ^b^	0.01 ± 0.002 ^C^	5.41 ± 0.28 ^a,b^	0.17 ± 0.05 ^A,B^	8.43 ± 1.41 ^a^	0.16 ± 0.01 ^B^
8	Unidentified carotenoid	22.18	nd	1.40 ± 0.18 ^a^	nd	2.13 ± 0.07 ^a^	nd	1.27 ± 0.02 ^b^	nd	1.15 ± 0.03 ^b^	nd	1.76 ± 0.41 ^a^	nd
9	Unidentified carotenoid	25.89	nq	1.21 ± 0.15 ^a^	nd	0.70 ± 0.01 ^b^	nd	1.37 ± 0.17 ^a^	nd	0.26 ± 0.08 ^b^	nd	nd	nd
10	Echinenone	27.01	nd	1.17 ± 0.07	nd	nd	nd	nd	nd	nd	nd	nd	nd
11	Unidentified carotenoid	28.93	nd	1.10 ± 0.04 ^b^	nd	nd	nd	2.02 ± 0.24 ^a^	nd	1.15 ± 0.14 ^b^	nd	1.00 ± 0.08 ^b^	nd
12	Unidentified carotenoid	29.7	nq	0.41 ± 0.16 ^b^	nq	1.15 ± 0.07 ^a^	nd	nd	nd	0.26 ± 0.09 ^b^	nd	nd	nq
13	β−Carotene	34.32	0.88 ± 0.11 ^B^	19.58 ± 0.64 ^c^	1.22 ± 0.11 ^A^	25.13 ± 0.75 ^b,c^	0.05 ± 0.002 ^D^	47.89 ± 2.81 ^a^	0.10 ± 0.002 ^D^	29.30 ± 0.56 ^b^	0.54 ± 0.05 ^C^	30.57 ± 1.63 ^b^	0.68 ± 0.01 ^B,C^
14	Phaeophytin-*a*	34.69	nq	5.81 ± 1.08 ^a^	nd	2.25 ± 0.61 ^b^	nd	1.81 ± 0.11 ^b^	nd	5.23 ± 0.27 ^a^	nd	6.31 ± 0.33 ^a^	nd
15	Unidentified carotenoid	35.05	nq	3.92 ± 0.18 ^c^	nq	5.23 ± 0.64 ^a,b^	nq	6.51 ± 0.24 ^a^	nq	3.37 ± 0.11 ^c^	nq	4.53 ± 0.60 ^b,c^	nq
16	γ−Carotene	41.78	nd	nq	nd	35.96 ± 3.00 ^a^	nd	33.52 ± 3.79 ^a^	nd	32.11 ± 1.47 ^a^	nd	26.51 ± 1.99 ^b^	nd
**Total carotenoids**			3.00 ±≤ 0.01 ^A^	115.98 ± 0.48 ^c^	1.55 ±≤ 0.01 ^B^	123.26 ± 0.85 ^b^	0.05 ±≤ 0.01 ^F^	150.15 ± 1.40 ^a^	0.18 ±≤ 0.01 ^E^	104.71 ± 0.42 ^d^	0.71 ±≤ 0.01 ^D^	113.81 ± 0.81 ^c^	0.84 ±≤ 0.01 ^C^
**Total chlorophylls**			nd	25.27 ± 1.17 ^a,b^	14.77 ± 1.21 ^A^	29.23 ± 0.69 ^a,b^	0.81 ± 0.06 ^C^	34.81 ± 1.52 ^a^	nd	12.69 ± 2.04 ^c^	9.42 ± 1.01 ^B^	21.90 ± 3.55 ^b,c^	8.69 ± 0.07 ^B^

^1^ Values are expressed as mean ± SD of two determinations. ^2^ Different superscript letters in the same row denote statistical differences at *p* < 0.05 (ANOVA, Tukey’s HSD); uppercase, Aqueouse extracts; lowercase, Acetonic extracts. nd: Not detected. nq: Not quantified.

**Table 2 marinedrugs-20-00761-t002:** Total Phenolic Content (TPC) and Total Proteins present in cyanobacterial extracts ^1,2^.

Strains	TPC (μg (GAE) mg^−1^ _dry extract_)		Total Proteins (μg (BSA) mg^−1^_dry extract_)	
	Aqueous Extracts	Acetonic Extracts	Aqueous Extracts	Acetonic Extracts
*Cyanobium* sp. LEGE 07175	14.79 ^a,b^ ± 0.20	23.11 ^a,b^ ± 0.69	309.63 ^e^ ± 4.26	167.49 ^b^ ± 4.93
Synechococcales cyanobacterium LEGE 181150	14.49 ^a,b^ ± 0.48	20.14 ^b,c^ ± 0.16	695.25 ^b^ ± 10.35	199.37 ^a^ ± 6.38
*Leptothoe* sp. LEGE 181156	14.93 ^a,b^ ± 0.27	27.13 ^a^ ± 0.07	463.70 ^d^ ± 8.78	158.67 ± 2.58
Synechococcales cyanobacterium LEGE 181157	14.92 ^a,b^ ± 0.66	17.86 ^c^ ± 0.06	760.81 ^a^ ± 15.72	128.84 ^c^ ± 7.12
Synechococcales cyanobacterium LEGE 181158	15.67 ^a^ ± 0.23	24.71 ^a,b^ ± 0.53	579.99 ^c^ ± 12.36	209.31 ^a^ ± 4.36
Oscillatoriales cyanobacterium LEGE 181159	11.46 ^b^ ± 0.67	−	550.59 ^c^ ± 18.14	−

^1^ Mean ± SD of three independent experiments. ^2^ Different superscript letters in the same column correspond to statistical differences at *p* < 0.05 (ANOVA, Tukey’s HSD). − not evaluated.

**Table 3 marinedrugs-20-00761-t003:** Phycobiliproteins content (μg mg^−1^_dry extract_) in cyanobacteria aqueous extracts ^1,2^.

Strains	*Phycobiliproteins* (μg mg^−1^ _dry aqueous extract_)		
	*Phycocyanin*	*Allophycocyanin*	*Phycoerythrin*
*Cyanobium* sp. LEGE 07175	18.21 ^f^ ± 1.36	11.26 ^e^ ± 1.20	8.06 ^e^ ± 1.17
Synechococcales cyanobacterium LEGE 181150	222.76 ^a^ ± 0.79	57.29 ^b^ ± 0.17	17.31 ^d^ ± 0.11
*Leptothoe* sp. LEGE 181156	76.10 ^e^ ± 0.13	37.05 ^d^ ± 0.28	159.05 ^b^ ± 0.14
Synechococcales cyanobacterium LEGE 181157	117.94 ^d^ ± 0.12	44.57 ^c^ ± 0.28	275.04 ^a^ ± 0.30
Synechococcales cyanobacterium LEGE 181158	194.43 ^c^ ± 0.39	44.52 ^c^ ± 0.34	17.42 ^d^ ± 0.22
Oscillatoriales cyanobacterium LEGE 181159	201.15 ^b^ ± 0.78	65.90 ^a^ ± 0.67	26.33 ^c^ ± 0.16

^1^ Values are expressed as the mean ± SD of three independent experiments. ^2^ Different superscript letters in the same column correspond to statistical differences at *p* < 0.05.

**Table 4 marinedrugs-20-00761-t004:** Inhibitory concentration (IC) values (μg mL^−1^) of cyanobacteria aqueous and acetonic extracts against superoxide anion radical (O_2_^•−^) ^1,2^.

Strains	O_2_^•−^ (μg mL^−1^)			
	Aqueous Extracts		Acetonic Extracts	
	IC_25_	IC_50_	IC_25_	IC_50_
*Cyanobium* sp. LEGE 07175	248.31 ^c^ ± 1.97	816.22 ^c^ ± 11.48	921.46 ^b^ ± 77.09	1549.10 ^b^ ± 65.28
Synechococcales cyanobacterium LEGE 181150	25.48 ^a,b^ ± 9.46	112.18 ^a^ ± 12.46	538.01 ^a^ ± 67.00	943.45 ^a^ ± 45.48
*Leptothoe* sp. LEGE 181156	41.10 ^a,b^ ± 5.24	nd	647.24 ^a^ ± 8.95	1097.05 ^a^ ± 2.41
Synechococcales cyanobacterium LEGE 181157	25.48 ^a,b^ ± 6.81	63.24 ^a^ ± 24.67	467.26 ^a^ ± 41.66	847.65 ^a^ ± 30.70
Synechococcales cyanobacterium LEGE 181158	51.22 ^b^ ± 24.20	404.59 ^b^ ± 14.30	1032.15 ^b^ ± 139.10	1463.69 ^b^ ± 271.87
Oscillatoriales cyanobacterium LEGE 181159	12.27 ^a^ ± 0.70	332.40 ^b^ ± 77.41	−	−

^1^ Mean ± SD of at least three independent experiments, performed in duplicate. ^2^ Different superscript letters in the same column correspond to statistical differences at *p* < 0.05 (ANOVA; Tukey’s HSD). nd: not determined. −: not evaluated.

**Table 5 marinedrugs-20-00761-t005:** Inhibitory concentration (IC) values (μg mL^−1^) of cyanobacteria aqueous and acetonic extracts obtained for hyaluronidase, elastase, and tyrosinase ^1,2^.

Strains	Hyaluronidase (μg mL^−1^)				Elastase (μg mL^−1^)		Tyrosinase (μg mL^−1^)	
	Aqueous Extracts		Acetonic Extracts		Acetonic Extracts		Acetonic Extracts	
	IC_25_	IC_50_	IC_25_	IC_50_	IC_25_	IC_50_	IC_25_	IC_50_
*Cyanobium* sp. LEGE 07175	488.39 ^a,b^ ± 97.20	894.59 ^c^ ± 78.14	796.94 ^a^ ± 284.55	nd	nd	nd	795.57 ^b^ ± 198.79	nd
Synechococcales cyanobacterium LEGE 181150	257.05 ^a,b^ ± 1.39	483.86 ^a^ ± 86.71	595.11 ^a^ ± 4.77	726.29 ^a^ ± 16.56	nd	nd	nd	nd
*Leptothoe* sp. LEGE 181156	800.00 ^c^ ± 152.73	nd	nd	nd	378.26 ^b^ ± 24.50	nd	133.47 ^a^ ± 35.40	465.92 ^a^ ± 37.59
Synechococcales cyanobacterium LEGE 181157	188.89 ^a^ ± 86.14	645.06 ^a,b^ ± 118.73	613.47 ^a^ ± 82.241	859.83 ^a^ ± 122.08	51.43 ^a^ ± 20.02	380.50 ± 19.72	381.33 ^a^ ± 14.378	849.48 ^b^ ± 75.815
Synechococcales cyanobacterium LEGE 181158	397.72 ^a,b^ ± 70.00	624.51 ^a,b^ ± 19.84	442.63 ^a^ ± 36.54	738.88 ^a^ ± 83.79	233.47 ^b^ ± 89.26	nd	nd	nd
Oscillatoriales cyanobacterium LEGE 181159	518.69 ^b,c^ ± 15.68	801.30 ^b,c^ ± 116.04	−	−	−	−	−	−

^1^ Mean ± SD of at least three independent experiments, performed in duplicate. ^2^ Different superscript letters in the same column correspond to statistical differences at *p* < 0.05 (ANOVA; Tukey’s HSD). nd: not determined. − not evaluated.

**Table 6 marinedrugs-20-00761-t006:** Percentage (%) of inhibition of collagenase displayed by cyanobacteria aqueous and acetonic extracts ^1,2^.

Strains	% Collagenase Inhibition			
	Aqueous Extracts		Acetonic Extracts	
	0.75 mg mL^−1^	1 mg mL^−1^	0.75 mg mL^−1^	1 mg mL^−1^
*Cyanobium* sp. LEGE 07175	nd	8.91 ^b^ ± 2.1	32.88 ^a^ ± 1.70	nd
Synechococcales cyanobacterium LEGE 181150	nd	nd	9.81 ^b^ ± 2.28	44.40 ^a^ ± 3.87
*Leptothoe* sp. LEGE 181156	nd	nd	36.61 ^a^ ± 2.72	36.74 ^a,b^ ± 7.82
Synechococcales cyanobacterium LEGE 181157	nd	nd	nd	nd
Synechococcales cyanobacterium LEGE 181158	nd	25.97 ^a^ ± 6.22	7.30 ^b^ ± 1.58	24.82 ^b^ ± 4.01
Oscillatoriales cyanobacterium LEGE 181159	nd	nd	−	−

^1^ Mean ± SD of at least three independent experiments, performed in duplicate. ^2^ Different superscript letters in the same column correspond to statistical differences at *p* < 0.05 (unpaired t-test; ANOVA; Tukey’s HSD). nd: not determined. − not evaluated.

**Table 7 marinedrugs-20-00761-t007:** Yield (%, *w/w*) obtained from sequential acetonic and aqueous extractions ^1^.

Strains	Solvents	
	Acetone	Water
*Cyanobium* sp. LEGE 07175	1.2 ± 0.3	17.2 ± 1.4
Synechococcales cyanobacterium LEGE 181150	1.4 ± 0.4	39.0 ± 5.2
*Leptothoe* sp. LEGE 181156	1.9 ± 0.6	26.6 ± 5.3
Synechococcales cyanobacterium LEGE 181157	1.6 ± 0.6	31.7 ± 8.4
Synechococcales cyanobacterium LEGE 181158	1.5 ± 0.3	23.4 ± 5.6
Oscillatoriales cyanobacterium LEGE 181159	−	38.9 ± 1.0

^1^ Values are expressed as Mean ± SD of at least four extractions. − not evaluated.

**Table 8 marinedrugs-20-00761-t008:** Calibration curves of authentic standards used for quantification of different carotenoids and chlorophylls.

Standards	Calibration Curve	*r* ^2^
Lutein	y = 31,188,975x + 81,368	0.9987
Chlorophyll-*a*	y =5,647,422x + 14,838	0.9989
Zeaxanthin	y = 40,108,171x + 97,810	0.9994
Cantaxanthin	y = 39,997,059x + 197,655	0.9992
Myxoxantophyll	y = 30,518,380x + 5976	0.9993
Echinenone	y = 74,770,292x + 126,878	0.9997
β-Carotene	y = 31,852,521x + 16,127	0.9999
γ-Carotene	y = 452,252x + 17,574	0.9913

## Data Availability

Data are available upon request to the corresponding author.

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
