# Peer review of "Cosmetic Application of Cyanobacteria Extracts with a Sustainable Vision to Skincare: Role in the Antioxidant and Antiaging Process"

_marinedrugs, 2022, doi:10.3390/md20120761_

Round 1

Reviewer 1 Report

To authors

Title Is ok

Introduction: Is ok

Results:

2.1. cytotoxicity, Pag 3, L 107-110, information should be in Materials and methods

2.2.4 Phycobiliproteins, pag 8, L 231-234, information should be not in this section.

Discussion:

3.1.2. Total phenolic content, pag 13, L 422-423, how do they explain that acetonic extract had higher phenolic content than aqueus extract, due to phenolic compound are more solubles in solvents of high polarity.

3.1.4 Phyconiliproteins, p 14, L 470-496,  the authors must search information on spirulina, since it is the main source of phycanin and compare with their results.

Materials and methods:

-- The procedures and techniques of Extract preparation, Total Proteins, Cell culture, Cytotoxicity – MTT assay, must have references

References:

-- References should be described in agreement with instructions for authors (full name of the authors

Author Response

First, we would like to thank the reviewers’ comments and suggestions which greatly contributed to improve our work.

  1. We appreciate the reviewer positive feedback on our title and introduction.
  2. Results:

2.1. cytotoxicity, Pag 3, L 107-110, information should be in Materials and methods

Our answer: The first paragraph, pag 3, L 107-110: “To assess the safety of the cyanobacteria acetonic and aqueous extracts, a preliminary in vitro cytotoxicity assay was performed using three different cell lines, the human keratinocytes HaCAT, the mice fibroblasts 3T3L1 and the human endothelial cells hCMEC/D3 at concentrations ranging from 12.5 to 200 mg mL-1.”, was placed in Material and methods section, pag. 18, L 713-716 as suggested. In both paragraphs the surrounding text was rewritten so that there is coherence.

2.2 Phycobiliproteins, pag 8, L 231-234, information should be not in this section.

Our answer: The text: “Phycobiliproteins are water-soluble proteins, deep-colored, formed by a complex between proteins and covalently bound phycobilins. Phycocyanin (PC), allophycocyanin (APC), and phycoerythrin (PE) are the three main PBPs groups, absorbing light within specific regions of the spectrum, according to their structure and types of bilins [9].” was placed to the discussion section in pag 13 L 474-477

  1. Discussion:

3.1.2. Total phenolic content, pag 13, L 422-423, how do they explain that acetonic extract had higher phenolic content than aqueous extract, due to phenolic compound are more solubles in solvents of high polarity.

Our answer: to explain the higher content of phenolic compounds in the acetonic extract the following text was added: “Although phenolic compounds are soluble in solvents with higher polarity, the fact that the extraction was carried out sequentially with acetone and water, lead to the extraction of most of the compounds soon with acetone.”

3.1.4 Phyconiliproteins, p 14, L 470-496, the authors must search information on spirulina, since it is the main source of phycanin and compare with their results.

Our answer: We agree that the genus Spirulina has been identified as one of the main producers of phycocyanin. In the text we added the phrase “For PC, the value described is within the concentrations obtained with different strains of Spirulina as exposed in a the recent review by Jaeschke and co-workers where different methods of PC extraction from Spirulina strains are compared” by using as a reference of a current review where a compilation of different methods of PC extraction in spirulina is made. In general, the values presented in the work are similar to those obtained by us.

  1. Materials and methods:

The procedures and techniques of Extract preparation, Total Proteins, Cell culture, Cytotoxicity – MTT assay, must have references

Our answer: references were added in the technics of material and methods that were asked. In the section “Extract preparation” The reference: Favas, R., Morone, J., Martins, R., Vasconcelos, V., & Lopes, G. (2022). Cyanobacteria Secondary Metabolites as Biotechnological Ingredients in Natural Anti-Aging Cosmetics: Potential to Overcome Hyperpigmentation, Loss of Skin Density and UV Radiation-Deleterious Effects. Marine Drugs20(3), 183. http://dx.doi.org/10.3390/md20030183 was added; in the section “Total proteins” the reference Favas, R., Morone, J., Martins, R., Vasconcelos, V., & Lopes, G. (2022). Cyanobacteria Secondary Metabolites as Biotechnological Ingredients in Natural Anti-Aging Cosmetics: Potential to Overcome Hyperpigmentation, Loss of Skin Density and UV Radiation-Deleterious Effects. Marine Drugs20(3), 183. http://dx.doi.org/10.3390/md20030183 was added; in the section “Cell culture” the references Morone, J.; Lopes, G.; Preto, M.; Vasconcelos, V.; Martins, R. Exploitation of Filamentous and Picoplanktonic Cyanobacteria for Cosmetic Applications: Potential to Improve Skin Structure and Preserve Dermal Matrix Components. Mar. Drugs 2020, 18, 486. https://doi.org/10.3390/md18090486 and Favas, R., Morone, J., Martins, R., Vasconcelos, V., & Lopes, G. (2022). Cyanobacteria Secondary Metabolites as Biotechnological Ingredients in Natural Anti-Aging Cosmetics: Potential to Overcome Hyperpigmentation, Loss of Skin Density and UV Radiation-Deleterious Effects. Marine Drugs20(3), 183. http://dx.doi.org/10.3390/md20030183 were added; in the “Cytotoxicity-MTT assay” a reference was already provided in the original version of the manuscript.

  1. References: References should be described in agreement with instructions for authors (full name of the authors

Our answer: References were corrected and are now described in agreement with instructions for authors.

Reviewer 2 Report

Τhis is an interesting work with potential applications for the cosmetic industry.

Some comments:

What was the reasoning behind the choice of the specific cell lines for the MTT assays? Why didn't you choose human fibroblasts, which is the norm for cosmetic applications?

The enzymic inhibition results are interesting but they rarely have a correspondence with the reality in a living cell. Why didn't you assess MMP activity in living cells?

You refer to sustainability in Line 832. Isn't the process of acetone extraction (and use of acetone) not sustainable? An LCA analysis, even for the lab/pilot scale would prove that.

Also please check for grammar/spelling/font errors, 

eg. Line 14 ...follow a every year growing...

Line 36 ...750 g 36 mL-1dry extract - Same for Line 526

Line 82 ... cyanobacterium was found to inhibits

Line 338 ... The absence of citotoxicity to

Author Response

First, we would like to thank the reviewers’ comments and suggestions which greatly contributed to improve our work.

1. Τhis is an interesting work with potential applications for the cosmetic industry.

Our answer: we appreciate the reviewer positive feedback on our paper.

2. Some comments:

What was the reasoning behind the choice of the specific cell lines for the MTT assays?

Our answer: keratinocytes are main epidermal cells and the first barrier to exogenous aggressors such as UV radiation. Fibroblasts are main dermal cells and responsible to produce most of the components of the extracellular matrix including collagen and elastin. Endothelial cells ate the typical cells of blood vessels epithelium and skin is a very vascularized organ in which blood supply is important for its normal physiology and regeneration. Therefore, components of a cosmetic formulation should not be toxic to keratinocytes, fibroblasts, and endothelial cells. On the other hand, the proliferation of these cells can be interesting in terms of, for example, cell regeneration.

Why didn't you choose human fibroblasts, which is the norm for cosmetic applications?

Our answer: we agreed that we should rather have used human fibroblasts. We do not use human fibroblasts because in the laboratory we only have the 3T3L1 strain and we had difficulty obtaining human strains. However, there are several works in the literature in which the 3T3L1 lineage is used as a fibroblast model also in human trials, so its use does not seem inappropriate.

The enzymic inhibition results are interesting but they rarely have a correspondence with the reality in a living cell. Why didn't you assess MMP activity in living cells?

Our answer: we are aware of the importance of measuring the activity of metalloproteinases in living cells. However, it seems to us that studying enzyme inhibition first restricts the number of strains that can really be of interest. In the most promising strains, we intend to continue with the study, namely by studying the inhibition of MMP in cells and the mechanisms involved.

 You refer to sustainability in Line 832. Isn't the process of acetone extraction (and use of acetone) not sustainable? An LCA analysis, even for the lab/pilot scale would prove that.

Our answer: We understand the reviewer's concerns about sustainability. In fact, all the acetone used in the extraction process is reused. During the execution of the acetone extracts, after the acetone evaporation, this solvent is collected and used again for new extractions. In addition, the use of acetone enables a quick and easy evaporation of the biomass. This method makes biomass available for subsequent water extraction. In this way, the biomass becomes more profitable, allowing the obtaining of natural extracts with different bioactivities, from the same biomass.

3. Also please check for grammar/spelling/font errors

 Our answer:  An extensive proofreading has been done throughout the entire text to correct grammatical and spelling errors. Specifically:

Line 14 ...follow a every year growing...

Our answer: we agree that this sentence is not illustrative of what is intended, so we replaced it with “Nature-based and sustainably sourced cosmetics has been dominating the area of skin care products”.

Line 36 ...750 g 36 mL-1dry extract - Same for Line 526

Our answer: The nomenclature has been standardized throughout the document.

Line 82 ... cyanobacterium was found to inhibits

Our answer: the sentence was corrected to “…cyanobacterium was found to inhibit…”. Also, an extensive proofreading has been done throughout the entire text to correct grammatical and spelling errors.

Line 338 … The absence of cytotoxicity to

Our answer: the sentence as corrected to “…the absence of cytotoxicity…”

Reviewer 3 Report

The manuscript is very interesting and presents the results of many studies conducted by the Authors. The following analyses of cyanobacteria extracts were performed: carotenoid and chlorophyll contents,  total phenolic content, total protein content, phycobiliproteins content, superoxide anion radical scavenging capacity, inhibition of hyaluronidase, elastase, and tyrosinase, as well as cytotoxic effects on the human keratinocytes HaCAT, the mice fibroblasts 3T3L1 and the human endothelial cells hCMEC/D3. The research has been well-planned and the results are properly presented.

However, in my opinion, the manuscript should be improved according to the following suggestions:

1.      The Authors should present the results of the cytotoxicity analysis of the extracts as a figure or table.

2.      The manuscript should include an explanation of why acetone extracts of Oscillatoriales 217 cyanobacterium LEGE 1181159 were not studied.

3.      Relating the data on superoxide anion radical scavenging capacity to the content of protein in the Results section is unnecessary.

4.      The Authors should indicate exactly what concentrations/dilutions of extracts they used in performing the following analyses: cytotoxicity, superoxide anion radical scavenging capacity, inhibition of hyaluronidase, elastase, and tyrosinase.

Author Response

First, we would like to thank the reviewers’ comments and suggestions which greatly contributed to improve our work.

1. The manuscript is very interesting and presents the results of many studies conducted by the Authors. The following analyses of cyanobacteria extracts were performed: carotenoid and chlorophyll contents, total phenolic content, total protein content, phycobiliproteins content, superoxide anion radical scavenging capacity, inhibition of hyaluronidase, elastase, and tyrosinase, as well as cytotoxic effects on the human keratinocytes HaCAT, the mice fibroblasts 3T3L1 and the human endothelial cells hCMEC/D3. The research has been well-planned, and the results are properly presented.

Our answer: we appreciate the reviewer positive feedback on our paper.

2. The Authors should present the results of the cytotoxicity analysis of the extracts as a figure or table.

Our answer: As we have many cytotoxicity results, we chose to present them as supplementary material. However, if the reviewer feels that it makes more sense to present it in the body of the article, we will be happy to make the change.

3. The manuscript should include an explanation of why acetone extracts of Oscillatoriales 217 cyanobacterium LEGE 1181159 were not studied.

Our answer: the sentence “Also due to cytotoxic effects in all cell lines tested, the acetone extract of Oscillatoriales cyanobacterium LEGE 181159 was also excluded. Thus, only non-toxic extracts were furthered analysed.” Was added pag.10 line 346-348

4. Relating the data on superoxide anion radical scavenging capacity to the content of protein in the Results section is unnecessary.

Our answer: We agree with the reviewer's suggestion. The text referring to protein content and PE was removed from the Results section and added to the Discussion section pag.13, L 492-496.

5. The Authors should indicate exactly what concentrations/dilutions of extracts they used in performing the following analyses: cytotoxicity, superoxide anion radical scavenging capacity, inhibition of hyaluronidase, elastase, and tyrosinase.

Our answer: We apologize for the omission. Concentrations of extracts are now provided in the revised manuscript.

Round 2

Reviewer 3 Report

The Authors revised the manuscript according to the comments. However, I suggest that the Authors add an appropriate description of the figure in the supplementary materials, including the data on the studied cell line and extract, since at presently it is not known what data the individual graphs represent.
After making this correction, I recommend the manuscript for publication.

Author Response

First, we would like to thank the reviewer comments and suggestions which greatly contributed to improve our work.

  1. The Authors revised the manuscript according to the comments.

Our answer:   We appreciate the reviewer positive feedback to the revision of the manuscript.

  1. However, I suggest that the Authors add an appropriate description of the figure in the supplementary materials, including the data on the studied cell line and extract, since at presently it is not known what data the individual graphs represent.
    After making this correction, I recommend the manuscript for publication.

Our answer:   We now present the supplementary material in a single file. Results for each cell line are presented separately. For each strain, the figure corresponding to each strain and extract with the respective incubation times is presented. As an example, we present the legend of figure S1:

Figure S1: Keratinocyte (HaCat) viability after 24 and 48 h of incubation with cyanobacteria aqueous (a-g) and acetone (h-n) extracts. (a, h) Cyanobium sp. LEGE 07175, (b, i) Synechococcales cyanobacterium LEGE 181150, (c, j) Leptothoe sp. LEGE 181156, (d, k) Synechococcales cyanobacterium LEGE 181157, (e, l) Synechococcales cyanobacterium LEGE 181158, (f, m) Oscillatoriales cyanobacterium LEGE 181159 and (g, n) Leptothoe sp. LEGE 181155. Results are expressed as % of MTT reduction vs. the untreated control. DMSO (20%) represents the positive control. Results are expressed as mean ± SD of at least three independent assays, performed in quadruplicate. Statistical differences at *p<0.05, **p<0.01, ***p<0.001, ****p<0.0001 (One way ANOVA, Tuckey HSD multiple comparisons test).

In the body of the manuscript, in section 2. Results and 2.1. Cytotoxicity, reference to figures in the supplemental material, where cytotoxicity results can be consulted, has been added (Pag. 3 Lines 106-109).

We hope that this description meets the reviewer's expectations.